# Clostridium butyricum improves cognitive dysfunction in ICV-STZ-induced Alzheimer's disease mice via suppressing TLR4 signaling pathway through the gut-brain axis

Yunfang Su[1,2☯], Dahui Wang[3,4☯], Ningning Liu[1], Jiajia Yang[5], Ruiqin Sun[1], Zhenqiang Zhang[1]*

1 Henan Engineering Research Center for Prevention and Treatment of Major Chronic Diseases with Chinese Medicine, Academy of Chinese Medical Sciences, Henan University of Chinese Medicine, Zhengzhou, Henan Province, China, 2 The First Affiliated Hospital of Henan University of Chinese Medicine, Zhengzhou, Henan Province, China, 3 LiangShan College (LiShui) China, Lishui University, Lishui, Zhejiang Province, China, 4 Henan JinBaiHe Biotechnology Co. LTD, Nanjing, Jiangsu, China, 5 School of Acupuncture and Massage, Henan University of Chinese Medicine, Zhengzhou, Henan Province, China

☯ These authors contributed equally to this work.
* 13333719963@126.com

**Data Availability Statement:** All relevant data are within the paper and its Supporting Information files.

## Abstract

In recent years, the relationship between gut-brain axis and Alzheimer's disease (AD) attracted increasing attention. The aim of this study is to investigate the therapeutic effect of Clostridium butyricum (CB) on intraventricular injection of streptozotocin (ICV-STZ)-induced mice and the potential mechanisms. ICV-STZ mice were treated with CB by gavage for 21 consecutive days. The pharmacological effect of CB was assessed by behavior test, brain tissue H&E staining and tau protein phosphorylation levels of hippocampus tissues. The expression levels of TLR4, MYD88, NF-κB p65, TNF-α, iNOS, Occludin and ZO-1 in hippocampal and colonic tissues were detected by Western-blot method. 16S rRNA gene sequencing analysis was used to analyze the intestinal microbiota of mice. The results showed that CB improved the cognitive dysfunction of ICV-STZ mice, restored the structure and cell number of hippocampal and cortical neurons, decreased the protein levels of p$^{Ser404}$-tau protein in hippocampal tissues and TLR4, MYD88, NF-κB p65 and iNOS in hippocampal and colonic tissues, and increased the protein levels of Occludin and ZO-1 in colonic tissues. Meanwhile, CB reversed the changes of intestinal microbiota in AD mice. Therefore, the mechanisms of cognitive function and brain pathological changes in AD mice improved by CB may be related to the regulation of TLR4 signaling pathway and intestinal microbiota. This study supports the potential anti-AD effect of CB and initially revealed its pharmacological mechanism of CB, providing a theoretical basis for further clinical application of CB.

**Funding:** The work had been funded by research start-up funds (00104311-2022-1-1-26), Program for Innovative Research Team (in Science and Technology) in University of Henan Province (21IRTSTHN026), 'liangshan' research project in Lishui University (FGL202202).

**Competing interests:** The authors have declared that no competing interests exist.

# Introduction

Alzheimer's disease (AD) is an insidious, progressive neurodegenerative disease characterized by cognitive impairment, abnormal mental behavior and decreased social functioning. As the population ages, the prevalence of AD increases each year [1]. It is estimated that the number of people with AD will reach 131 million worldwide and the socio-economic cost will reach $9.12 trillion in 2050 [2]. Currently, there are limited drugs used for the clinical therapy of AD, which can only relieve disease symptoms but not stop the progression of the disease [3]. AD is a serious threat to human health and a hindrance to socio-economic development.

The pathogenesis of AD is complex with typical pathological features including extracellular senile plaques (SP) formed by β-amyloid protein (Aβ) in the brain, intracellular neurofibrillary tangle (neurofibrillary tangle, NFTs) caused by tau protein hyperphosphorylation, and progressive brain neuronal loss [4]. In recent years, studies have found that the pathogenesis of AD is closely related to the gut-brain axis [5, 6]. Intestinal microbiota imbalance is prevalent in AD patients [7] and also reported in various AD animal models, such as APP/PS1 transgenic mice [8] and 5×FAD transgenic mice [9]. The gut-brain axis is bidirectional information communication system that integrates brain and gut functions and relies mainly on neural, endocrine and immune pathways [10]. Neuroinflammation is a major pathological feature of AD and an important indicator for drug screening [11, 12]. Chronic intestinal inflammation further promotes AD progression through the gut-brain axis due to the presence of intestinal microbiota imbalance and intestinal barrier dysfunction in AD patients [7, 13–15]. Studies have shown that probiotic supplementation has a therapeutic effect [16, 17]. *Clostridium butyricum* (CB), a probiotic, has been reported to improve neuroinflammation in APP/PS1 transgenic AD mice [18]. Streptozotocin (STZ) is a neurotoxin which widely used to induce hyperphosphorylation of tau of AD pathology through intraventricular injection (ICV) [19–22]. However, it is not clear whether CB has beneficial effects on ICV-STZ-induced AD model mice. The aim of this study was to investigate whether CB ameliorates ICV-STZ-induced AD pathology and to explore the underlying mechanisms.

In this study, we investigated the efficacy of CB using ICV-STZ-induced AD mice. The expression levels of TLR4/MYD88/NF-κB signaling pathway-related proteins in hippocampal and colonic tissues were tested, the intestinal tight junction protein levels were examined and intestinal microbiota were analyzed to investigate the potential mechanisms of CB on AD. The results showed that CB has anti-AD effect and the mechanisms are related to the regulation of the TLR4 signaling pathway and intestinal microbiota.

# Materials and methods

## Materials

CB powder ($1×10^{10}$ CFU/g) was provided by Henan JinBaiHe Biotechnology Co. LTD. The main materials for this study include STZ (S0130, SIGMA), isoflurane (R510-22, Shenzhen Ruiwade Life Technology Co., LTD), Hematoxylin-Eosin Staining Kit (G1120, Solarbio), antibodies of β-Actin (BM0627, BOSTER), GAPDH (GB12002, Servicebio), $p^{Ser404}$-tau (20194T, Cell Signaling Technology), TLR4 (CST 14358S), MYD88 (CST 4283S), NF-κB p65 (GB11997, Servicebio), TNF-α (ab1793, Abcam), iNOS (13120S, Cell Signaling Technology), Occludin (GB111401, Servicebio), ZO-1 (GB111402, Servicebio), Horseradish peroxidase (HRP)-linked secondary antibodies against rabbit and mouse IgG of the primary antibodies (ZB-2301; ZB-2305, Beijing Zhong Shan-Golden Bridge Biological Technology) and Goat anti-Rabbit IgG (H+L) Cross-Adsorbed Secondary Antibody, Alexa Fluor™ 594 (A-11012, Thermo Fisher Scientific).

## Animals and drug treatment

Six- to eight-week-old male C57BL/6N mice (20–30 g) were purchased from Beijing Vital River Laboratory Animal Technology Co., Ltd. Experimental animal license number: SCXK (jing) 2021–0006. Animal housing environment: room temperature (22±2˚C, humidity 60%, light/dark cycle 12 h, free diet and water. The animal experimental design was in accordance with the requirements of the Experimental Animal Ethics Committee of Henan University of Traditional Chinese Medicine (DWLLGZR202202147).

After 1 week of adaptive feeding, 3 mg kg$^{-1}$ STZ was injected into the lateral ventricle to induce AD model mice. Firstly, STZ (30 μg μl$^{-1}$) was dissolved in sodium citrate (pH 4.2, 1% w/v) and prepared before injection. For sodium citrate buffer preparation, 2.1 g of citric acid (FW: 210.14) and 2.94 g of sodium citrate (FW: 294.10) was diluted in 100 mL of distilled water to form liquid A and liquid B, respectively. Liquid A and liquid B were mixed 1:1 to form sodium citrate buffer pre-use. Next, mice were fixed in a brain stereotaxic instrument. After anesthesia, the fontanelle was exposed through a median incision at the top of the lateral ventricle. The right lateral ventricle was slowly injected with STZ (3 mg kg$^{-1}$) for 5 min, staying for 5 min, and then slowly withdrawn. All operations were performed under aseptic conditions. Finally, the skin incision was treated with penicillin and the wound was sutured. The sham-control group (control group) was injected with an equal volume of sodium citrate buffer. A control group, a model group and a CB-treated group were set up with 10 mice of each. CB was dissolved in PBS to an optimized dose of 2×10$^8$ CFU/d pre-use [18]. Equal volumes of PBS were given to the control and model groups. Mice were treated by gavage for 21 consecutive days. The experimental design of this study is shown in Fig 1A.

## Behavior experiments

Behavioral experiments were performed 5 days before the end of the animal experiment. Morris water maze test: mice were placed in water facing the wall of the pool from four different entry points and the time it took to climb the platform (i.e., the escape latency) was recorded for 5 consecutive days. Finally, the platform was removed and the mice were placed in the water facing the wall of the pool from the same entry point. The target quadrant residence time and the number of times crossing the platform within 60 s were recorded. Y maze experiment: mice were placed at the intersection of three maze arms in turn and moved freely for 5 min. The system recorded the total number and order of mice entering the arms and calculated the accuracy of spontaneous alternation of mice. Open field experiment: the mice were allowed to move freely in the open field for 5 min, and the total distance of movement and number of grooming were recorded.

## Histological examination

Mice were anesthetized with isoflurane, and the brain tissues were collected after 4% paraformaldehyde systemic perfusion. Immunohistochemical (IHC) staining was used to compare the histopathological changes of mice between groups. All brain tissues were paraffin-embedded and routinely sectioned. 5 μm-thick tissue sections were prepared and stained with hematoxylin-eosin (H&E). A microscope (Axioscope 5, ZEISS, Germany) was used to observe and take tissue images. Brain histopathological changes of mice were observed.

## Immunofluorescence (IF) assay

Paraffin sections were dewaxed in xylene, rehydrated, treated with 0.5–1% protease, and blocked with 5% skim milk for 1.5 h. p$^{Ser404}$-tau was used as primary antibody, Alexa Fluor 594 goat anti-mouse IgG (H + L) (Thermo Fisher Scientific) was used as secondary antibody,

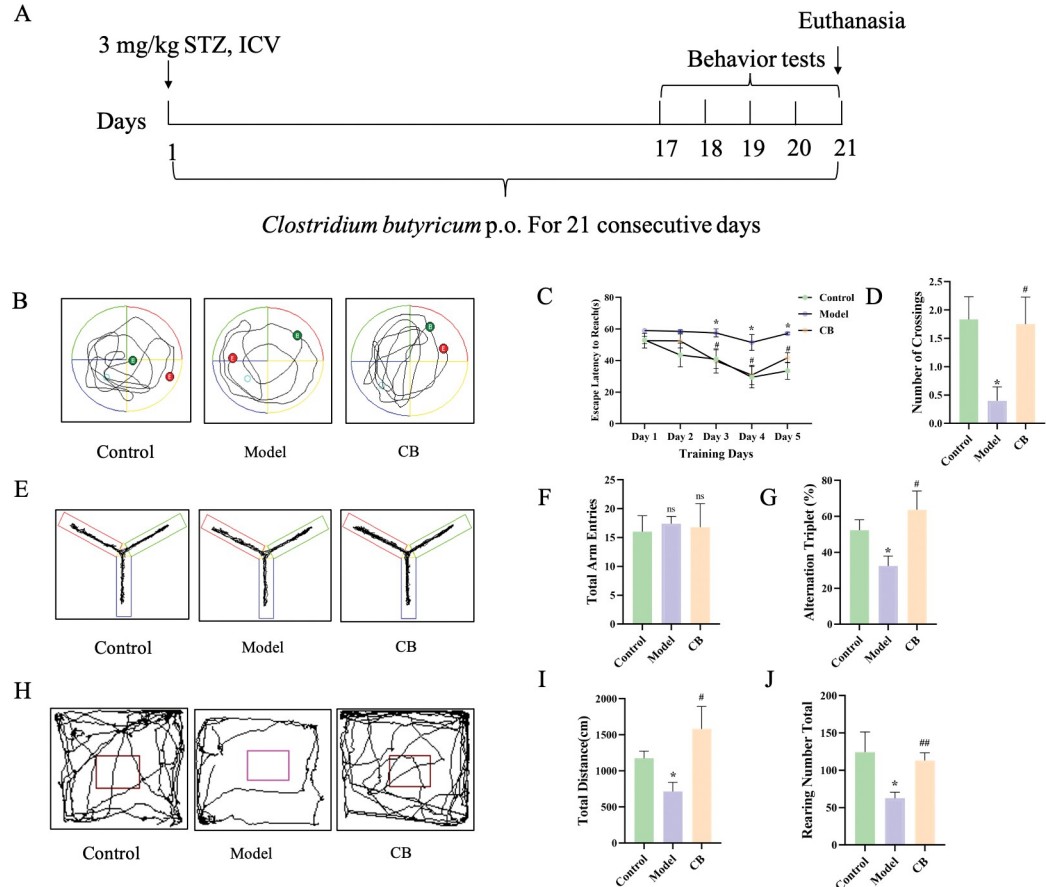

**Fig 1. Experimental design and behavioral experiments.** A. 3 mg kg$^{-1}$ STZ was injected into the lateral ventricle to induce AD model mice. Mice of control group were injected with an equal volume of sodium citrate buffer. Mice in CB-treated group were continuously treated with CB (2×10$^{8}$ CFU/d) by gavage for 21 consecutive days. The control and model groups were treated with equal volumes of PBS. Behavior tests were performed during the last 5 days. Tissues were collected after euthanasia. B. Morris water maze swimming trajectories of mice in each group; C. Morris water maze evasion latency; D. The number of platform crossings in Morris water maze of mice in each group; E. Y maze activity trajectory of mice in each group; F. The total number of Y maze arm entries in each group; G. Spontaneous alternating correct rate of Y maze in each group; H. Open field activity trajectories of mice in each group; I. Number of grooming; J. Total distance. * model group compared with the control group, # CB group compared with the model group.

and finally nuclear staining was performed with DAPI. A microscope (Axioscope 5, ZEISS, Germany) was used to observe and take tissue images.

## Western-blot assay

Appropriate amounts of tissue lysate were added to the hippocampal and colonic tissues of each group of mice, ground in a grinder at 4°C, lysed on ice for 30 min, and centrifuged at 4°C for 15 min at 12000 r min$^{-1}$. Total proteins were collected and protein concentrations were detected by BCA kit. Western-blot method was performed to detect the protein levels of p$^{Ser404}$-tau, TLR4, MYD88, NF-κB p65, TNF-α, iNOS, Occludin and ZO-1, and β-Actin or GAPDH was used as an internal control for data analysis.

## 16S rRNA gene sequencing of intestinal microbiota in cecal contents

The contents of the cecum were collected from 5 mice per group after anesthesia. All samples were immediately placed in sterile tubes and stored at -80°C until analysis. Bacterial DNA was

routinely extracted and primers were used to amplify the 16S rRNA gene V3-V4 variable region sequencing library (F: ACTCCTACGGGAGGCAGCA, R: GGACTACHVGGGTWTCTAAT), which was sequenced on Novaseq-PE250 (Illumina, Inc) platform. The amplicon sequence variants (ASVs)/operational taxonomic units (OUT) were clustered using the DADA2 method combined with the Vsearch method quality control to obtain the amplicon sequence variation (ASVs)/OUT abundance table with 100% similarity. Alpha diversity analysis (including chao1 index, shannon index, Plelou_e index, etc.) and species analysis (including species composition analysis and heat map analysis, etc.) were used to evaluate the abundance and diversity of the microbiota. Beta diversity analysis based on weighted UniFrac was used to evaluate the differences in microbiota structure of mice among groups (including principal coordinate analysis, PCOA analysis; non-metric multidimensional scaling, NMDS, etc.). 16S rRNA gene sequencing analysis was done by Suzhou PANOMIX Biomedical Tech Co., LTD [23].

## Statistical analysis

GraphPad Prism 9 software was used for statistical analysis. The experimental data were analyzed using one-way ANOVA and t-test. Data were expressed as M ± SEM, and $p < 0.05$ was considered significant.

## Results

### CB improves learning and memory and spatial exploration in AD mice

The results of the Morris water maze positioning navigation experiment showed that the escape latency of mice in each group gradually decreased with increasing training time and number. Compared with the control group, the evasion latency of the model group was significantly longer since day 3 ($p < 0.05$), and the evasion latency of the CB-treated group was significantly shorter since day 3 compared with the model group ($p < 0.05$) (Fig 1C). The longer the time elapsed in the target quadrant and the greater the number of crossings, the better the spatial learning and memory ability of mice. The results of the spatial exploration experiment showed that the number of platform crossing was significantly reduced in the model group compared with the control group ($p < 0.05$), and the number of platform crossing was significantly increased in the CB-treated group compared with the model group ($p < 0.05$) (Fig 1D). Y-maze test showed no significant difference in the total number of upper limb approaches in each group. Spontaneous alternation accuracy was significantly lower in the model group compared to the control group ($p < 0.05$), and significantly higher in the CB-treated group compared to the model group ($p < 0.05$) (Fig 1F, 1G). The high alternation rate means that the mice have better spatial exploration ability. The results of open field test showed that the number of grooming was significantly reduced in the model group compared to the control group ($p < 0.05$), while the number of grooming was significantly increased in CB-treated group compared to the model group ($p < 0.05$). The total distance of exercise was significantly lower in the model group than that in the control group ($p < 0.05$), and significantly increased in the CB-treated group ($p < 0.01$) (Fig 1I, 1J). The total distance moved and the number of grooming are the stress behaviors produced in response to a new and different environment which indicates that the mice are more excited and have better autonomous locomotor ability.

### CB improves brain pathology in AD mice

As shown in the Fig 2, neurons in hippocampal CA1, CA3, dentate gyrus (DG) and cortical areas of the control group were structurally intact, neatly arranged and with clear nuclear staining, whereas neuronal cells in the model group were loosely arranged with karyopyknosis

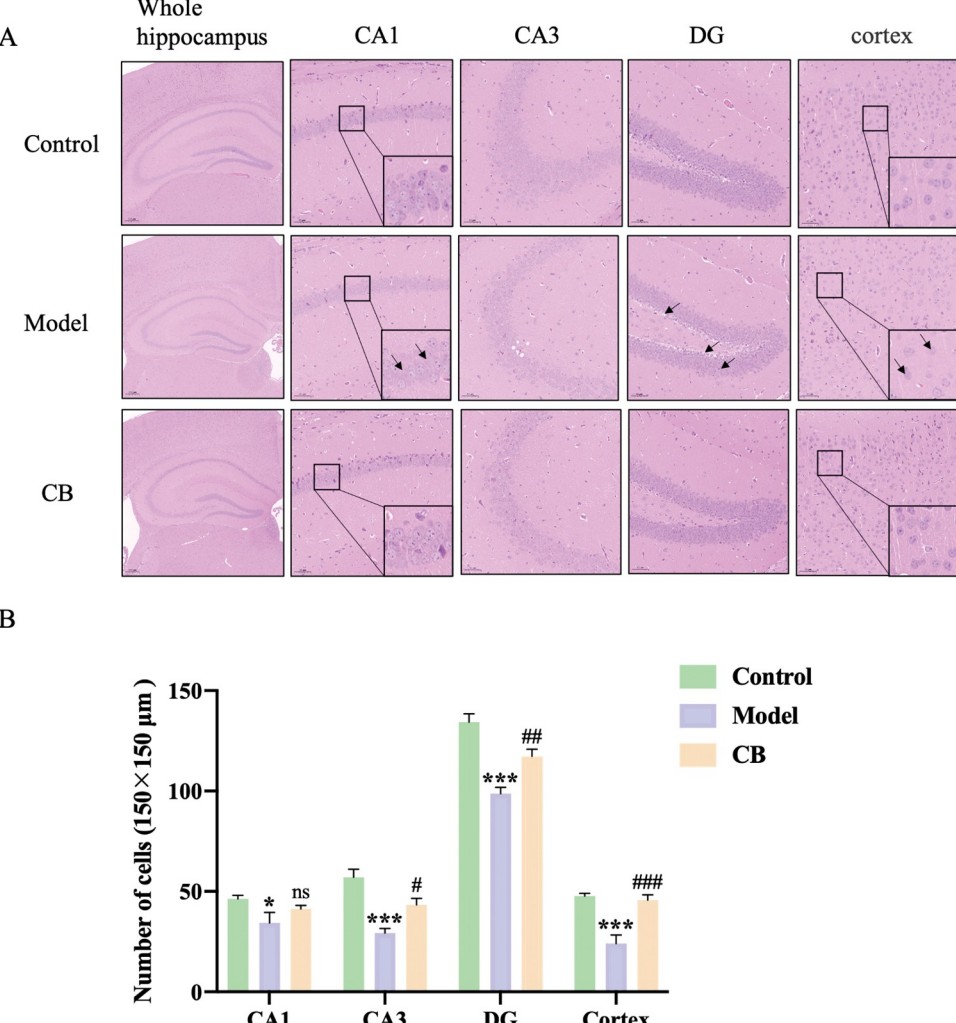

**Fig 2. H&E staining of brain tissues of mice in each group.** The whole hippocampus is Magnification 40×; CA1, CA3, cortex is Magnification 200×, partial enlarged in CA1 and cortex is Magnification 1400×. A. Histology of brain tissue in control group of mice was normal. Neuronal cells were loosely arranged with karyopyknosis (as shown by the black arrow). CB-treated group had a more regular arrangement of neuronal cells. B. Compared with the control group, the number of neuronal cells in hippocampus and cortex of mice in model group was significantly reduced ($p < 0.05$, $p < 0.001$, $p < 0.001$, $p < 0.001$), while the number of neuronal cells in the CB-treated group was increased compared to the model group ($p < 0.05$, $p < 0.01$, $p < 0.001$). Each vertical line represents the mean ± SEM of 3 identical non-overlapping regions in different groups (150×150 μm). * model group compared with the control group, # CB group compared with the model group.

(as shown by the black arrow) and the number of neuronal cells was significantly reduced ($p < 0.05$, $p < 0.001$, $p < 0.001$, $p < 0.001$) (Fig 2B). Compared with the model group, CB-treated mice showed improved pathological changes in hippocampus and cortex with more regular arrangement and significantly increased number of neuronal cells in CA3, DG and cortical areas ($p < 0.05$, $p < 0.01$, $p < 0.001$).

## CB inhibits hippocampal tau protein phosphorylation in AD mice

As shown in Fig 3A, 3B, by IF staining for the $p^{Ser404}$-tau protein, strong fluorescent signal intensity was observed in the brain tissues of the model mice, while weak fluorescence signal

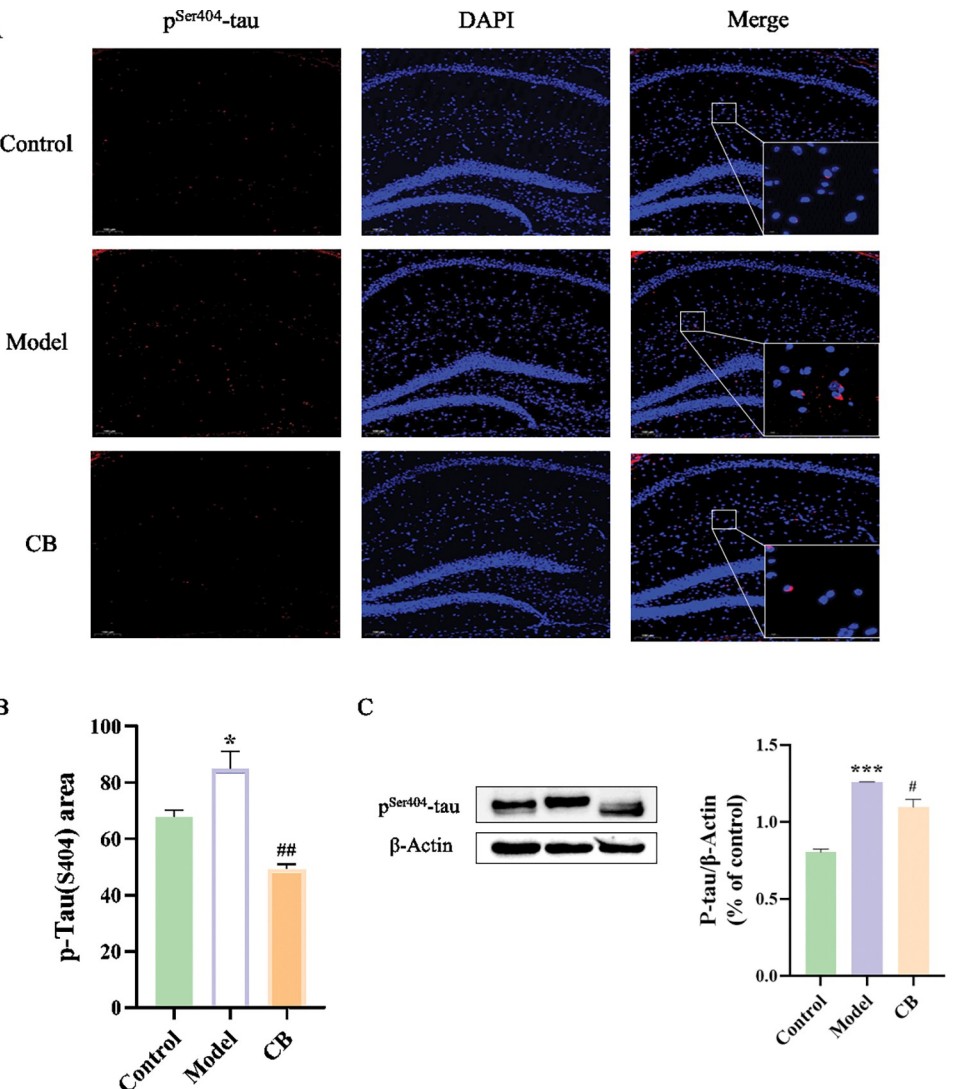

**Fig 3. IF staining and Western-blot of p$^{Ser404}$-tau protein in mouse hippocampal tissues.** A. IF staining of p$^{Ser404}$-tau protein (red) and nuclei re-stained (blue) (Magnification 100×, partial enlarged is Magnification 1000×); B. Expression levels of p$^{Ser404}$-tau in hippocampal tissues. Each vertical line represents the mean ± SEM of 3 mice per group; * significantly different from the control group, # significantly different from model group using one-way ANOVA followed by Tukey's multiple comparisons test.

intensity was observed in the CB-treated mice (red, representing positive for p$^{Ser404}$-tau). Similarly, as shown in Fig 3C, western-blot results showed that the expression level of p$^{Ser404}$-tau protein was significantly increased in the model group compared to the control group ($p < 0.001$), and the protein levels of p$^{Ser404}$-tau was significantly decreased in the CB-treated group compared to the model group ($p < 0.05$).

## Effect of CB on protein expression of TLR4, MYD88, NF-κB p65, TNF-α, iNOS, Occludin and ZO-1 in hippocampus and colon tissues of AD mice

Figs 4 and 5 showed that the protein expression levels of TLR4, MYD88, NF-κB p65, TNF-α and iNOS were significantly increased in hippocampal and colonic tissues of mice in model group compared to the control group ($p < 0.001$), while the expression levels of these proteins

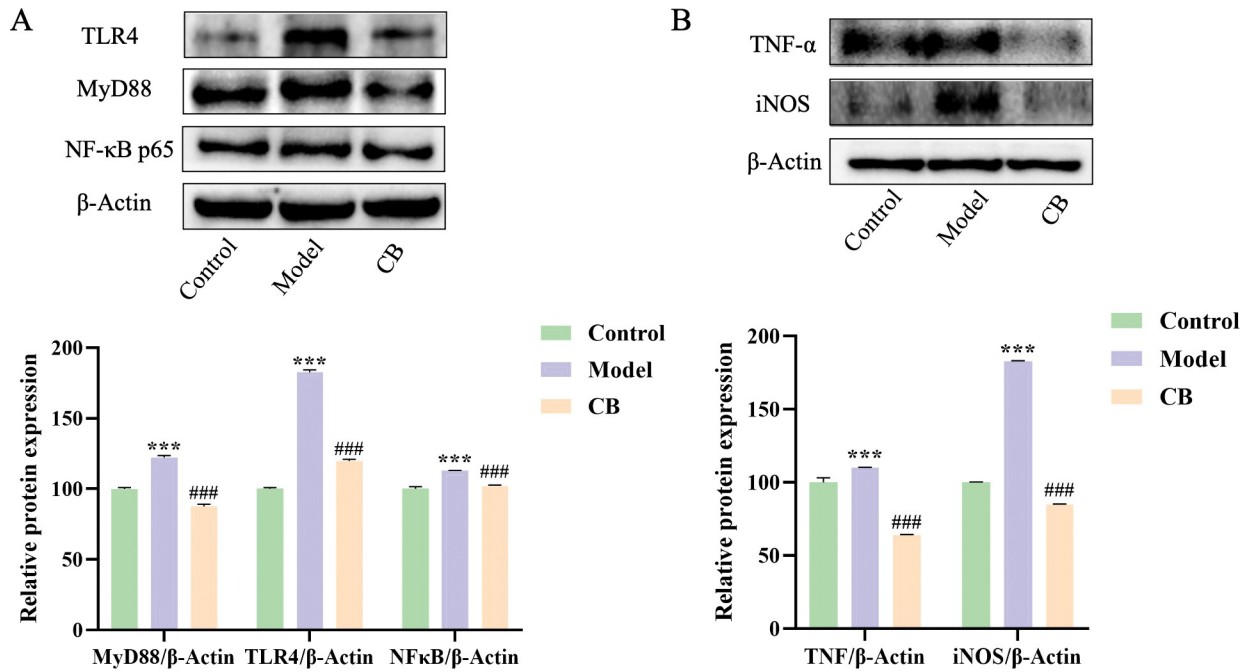

**Fig 4. Expression levels of TLR-4, MYD88, NF-κB P65, TNF-α and iNOS in hippocampal tissues of mice in each group.** Each vertical line represents the mean ± SEM of 3 mice per group; * significantly different from the control group, # significantly different from model group using one-way ANOVA followed by Tukey's multiple comparisons test.

were significantly decreased in the CB-treated group compared to the model group (p < 0.001). In addition, the levels of Occludin and ZO-1 were significantly decreased in colon tissues of mice in the model group compared to the control group (p < 0.001), while which were significantly increased in the CB-treated group (p < 0.001) (Fig 5C).

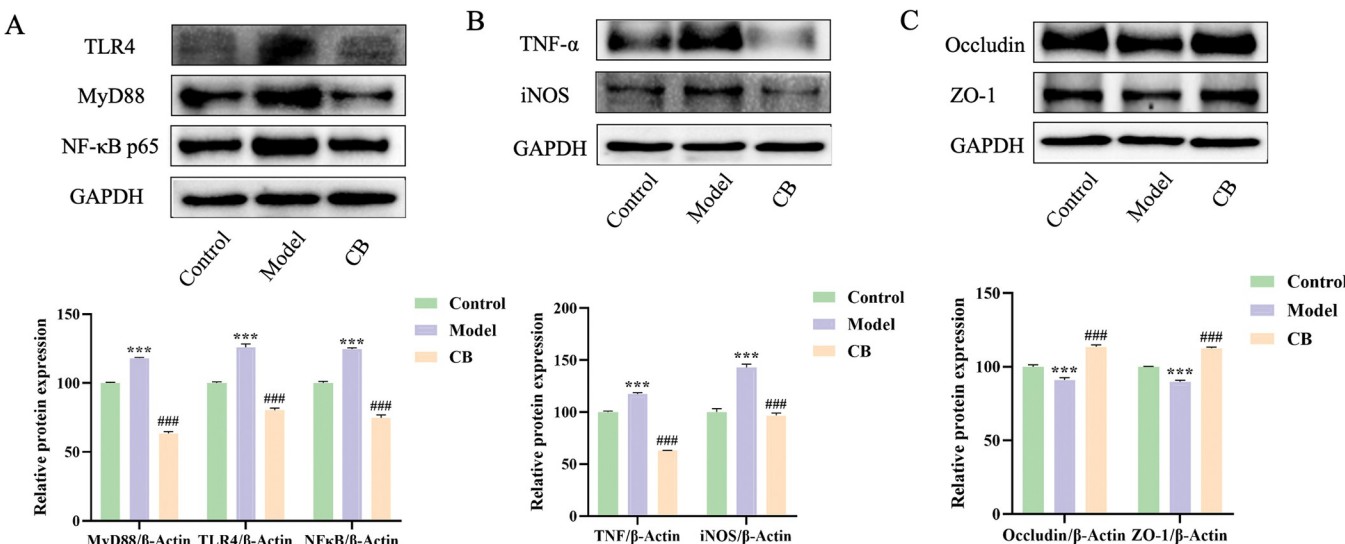

**Fig 5. Expression levels of TLR-4, MYD88, NF-κB P65, TNF-α, iNOS, Occludin and ZO-1 in colonic tissues of mice in each group.** Each vertical line represents the mean ± SEM of 3 mice per group; *significantly different from the control group, #significantly different from model group using one-way ANOVA followed by Tukey's multiple comparisons test.

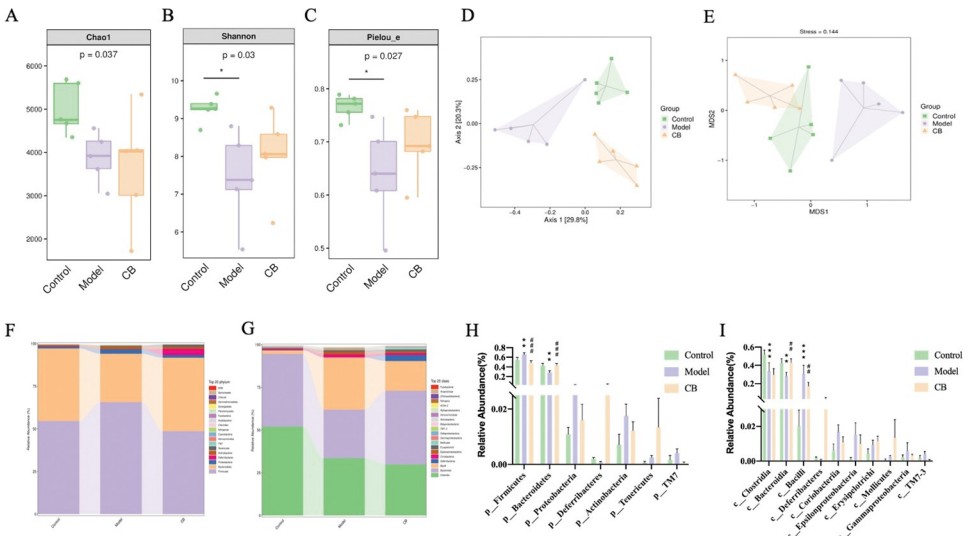

**Fig 6. Analysis of intestinal microbiota in cecal contents of mice in each group.** A-C. Chao1, Shannon and Plelou_e values of alpha diversity index in each group. D, E. PCOA and NMDS analysis of beta diversity; F-I. Significance analysis of species composition at the phylum, class levels. * significantly different from the control group, # significantly different from model group using one-way ANOVA followed by Tukey's multiple comparisons test.

## Effects of CB on intestinal microbiota

**Analysis of alpha and beta diversity.** As shown in Fig 6A–6C, alpha diversity showed that Chao1 values were decreased in the model group compared to the control group with no significant difference ($p > 0.05$), while Shannon and Plelou_e values were significantly decreased ($p < 0.05$). Chao1, Shannon and Plelou_e values were elevated in the CB-treated group were increased compared to the model group with no significant difference ($p > 0.05$). Beta diversity analysis showed that the groups were distributed in mutually independent areas, indicating differences in the structure of the microbiota of each group. NMDS analysis showed that the microbiota of mice in the CB-treated group partially overlapped with that of the control group compared to the model group, indicating that its microbiota composition was more inclined to that of the control group (Fig 6D, 6E).

**Species distribution of intestinal microbiota in each group.** Fig 6F–6I showed the significant differences in the abundance of ASVs/OTU sequences between different groups at the phylum and phylum levels. The results show a significant decrease in the abundance of Bacteroidetes, Clostridia and Bacteroidia ($p < 0.01$, $p < 0.001$), and a significant increase in the abundance of Firmicutes and Bacill ($p < 0.01$, $p < 0.001$). CB significantly reversed the changes in abundance of Bacteroidetes, Bacteroidia, Firmicutes and Bacill in AD mice ($p < 0.01$, $p < 0.001$).

## Discussion

In recent years, the relationship between brain-gut axis and neurological disorders has received increasing attention [6, 24]. The gut-brain axis describes the interactions between the intestinal epithelium, mucosal immune system and microbiota, and the enteric nervous system in the intestinal environment [10]. The immunopathogenesis of AD is considered to be related to the composition of gut microbiota [25] or even mediated by gut microbiota [26]. Gut microbiota imbalance is strongly associated with the development of AD [27]. Recent studies have found that targeting intestinal microbiota holds good anti-AD prospects [7, 28–30]. Probiotics

are commonly used to regulate gut health. Given the close relationship between gut-brain axis and AD, researchers have attempted to use probiotics to treat AD, which have been reported to alleviate AD through anti-oxidation, apoptosis, inflammation, and insulin resistance [21, 31]. CB is a common probiotic that typically exerts its therapeutic effects by regulating intestinal microbiota, intestinal barrier function, and inhibiting inflammation in the treatment of intestinal diseases. Recent study has reported that CB has protective effects against microglia-mediated neuroinflammation in APP/PS1 transgenic mice through modulation of intestinal microbiota and butyric acid metabolites [18]. ICV-STZ-induced mouse is a common model of sporadic AD which is valuable in the study of pathogenesis and drug research of AD. This study is the first to investigate the protective effects of CB in ICV-STZ-induced AD model mice, and to explore the potential mechanisms from the perspective of the gut-brain axis, providing a basis for further clinical applications of CB.

The results of behavioral tests indicated that CB improved cognitive dysfunction in ICV-STZ-induced AD mice, which is consistent with the reported results in APP/PS1 AD mice [18]. H&E staining showed reduced hippocampal pathological damage in CB-treated mice, suggesting a neuroprotective effect of CB in ICV-STZ-induced AD mice. Phosphorylated tau protein deposition is considered to be one of the markers of AD pathology [32, 33]. Western-blot and IF staining results showed that $p^{Ser404}$-tau protein levels were significantly reduced in the CB group, indicating that CB reduced ICV-STZ-induced AD mice hyperphosphorylated tau protein accumulation in brain and improved the pathology of AD.

TLR4 is a key component of the innate immune system which has been used as a potential target for AD therapy [34]. Modulation of TLR4/MYD88/NF-κB signaling pathway can improve neuroinflammation and cognitive function in LPS-induced and transgenic AD mice [35, 36]. In the present study, we found that CB significantly decreased the protein expression levels of TLR4, MYD88, NF-κB p65, TNF-α and iNOS in colon and brain tissues of AD mice, suggesting that CB may improve neuroinflammation by regulating the gut-brain axis. In addition, this study found that CB significantly increased the protein levels of Occludin and ZO-1 in colonic tissues of AD mice, suggesting that CB has a potential regulatory function on the intestinal barrier.

How the gut microbiota is altered in AD patients is still controversial [7]. In the present study, the alpha diversity Shannon and Plelou_e values were significantly lower in the model group, indicating a significant decrease in the abundance and diversity of the microbiota in AD mice. Beta diversity showed there are differences in the structure of the microbiota between the model and control group. Firmicutes and Bacteroidetes are the largest components of intestinal microbiota. After CB treatment, the intestinal microbiota structure of mice converged to healthy level and Bacteroidetes, Bacteroidia, Firmicutes and Bacill levels were reversed, suggesting that CB has a regulatory effect on intestinal microbiota structure in ICV-STZ-induced AD mice. The colonization of probiotics usually requires a period of time. Some studies have reported that probiotics colonization in humans occurs within 2–3 weeks, however, mice are resistant to probiotic colonization, and even then, low levels of probiotics colonization induce modulation of the lower gastrointestinal mucosal microbiome [37]. Therefore, in the present study, although the level of colonization may be low after 21 days of CB treatment, it is sufficient to affect the composition of the intestinal microbiota, as reported in a study of probiotic treatment of ICV-STZ-induced AD mice [21]. Studies have shown increased bacterial populations in brain tissue of AD patients compared to normal individuals [38]. Brain microbiological analysis will be performed in further studies to provide more evidence for CB in the treatment of AD.

In conclusion, CB ameliorates ICV-STZ-induced cognitive dysfunction in AD mice not only by modulating TLR4/MYD88/NF-κB signaling on the gut-brain axis, but also by regulating intestinal microbiota (Fig 7). This study provides a theoretical basis for the clinical

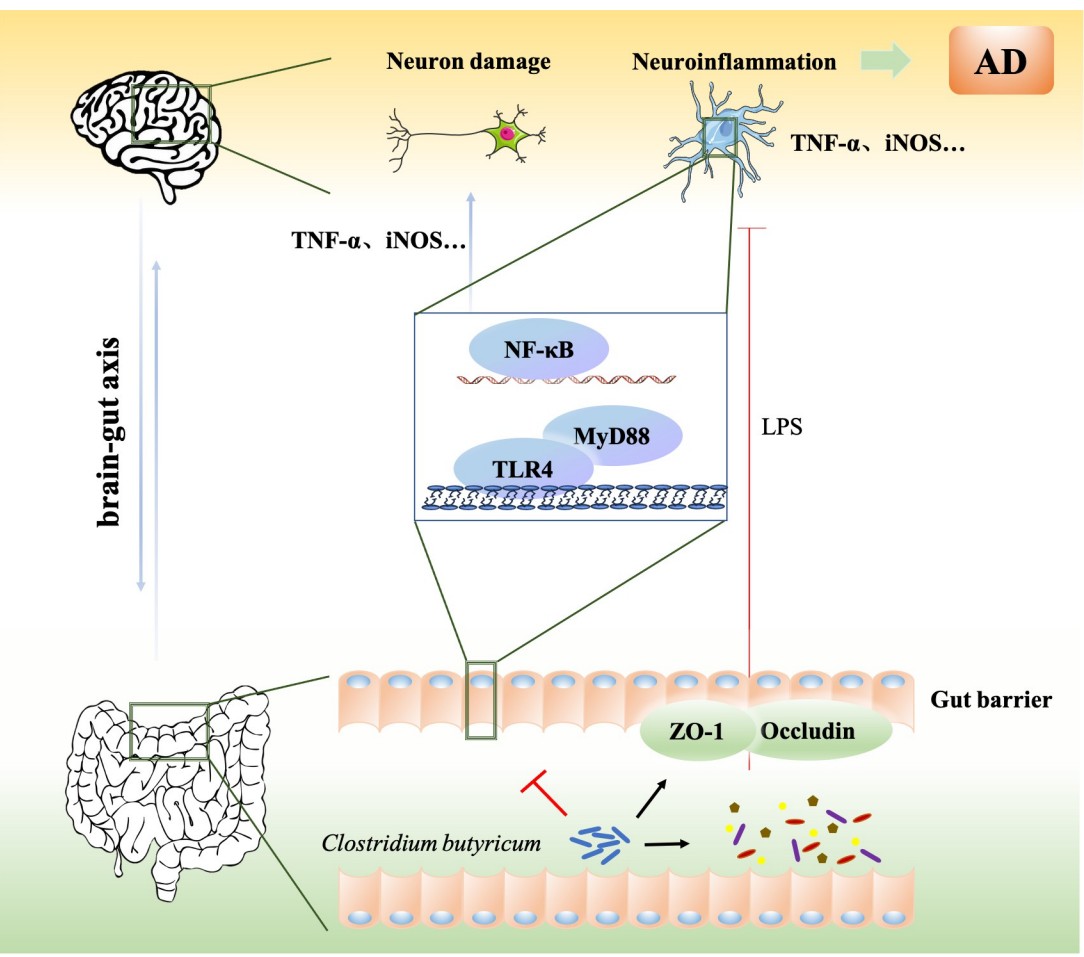

**Fig 7. Protective effects of CB in ICV-STZ-induced AD mice.** CB ameliorates ICV-STZ-induced cognitive dysfunction in AD mice by modulating the TLR4/MYD88/NF-κB signaling pathway through gut-brain axis.

**Table 1. Acronym comparison table.**

| Full name | Acronym |
| --- | --- |
| Alzheimer's disease | AD |
| Clostridium butyricum | CB |
| Intraventricular injection | ICV |
| Streptozotocin | STZ |
| Intraventricular injection of streptozotocin | ICV-STZ |
| Senile plaques | SP |
| β-amyloid protein | Aβ |
| Neurofibrillary tangle | NFTs |
| Immunohistochemical | IHC |
| Immunofluorescence | IF |
| Hematoxylin-eosin | H&E |
| Amplicon sequence variants | ASVs |
| Operational taxonomic units | OUT |
| Dentate gyrus | DG |

application of CB. In addition, this study is limited in that only one animal model of AD was selected for study without reverse validation. Further reverse validation of CB in AD is needed, such as intestinal microbiota transplantation.

Abbreviations of the full text are shown in Table 1.

## Supporting information

**S1 Raw images.**
(PDF)

**S1 File.**
(ZIP)

**S2 File.**
(ZIP)

**S3 File.**
(ZIP)

## Acknowledgments

*Clostridium butyricum* used in this study is kindly provided by Henan JinBaiHe Biotechnology Co. LTD.

## Author Contributions

**Conceptualization:** Ningning Liu, Zhenqiang Zhang.

**Data curation:** Yunfang Su, Dahui Wang, Jiajia Yang.

**Funding acquisition:** Yunfang Su, Dahui Wang, Zhenqiang Zhang.

**Writing – original draft:** Yunfang Su, Dahui Wang, Ningning Liu.

**Writing – review & editing:** Yunfang Su, Dahui Wang, Ruiqin Sun.

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
