## [Decision Letter · Decision Letter 0]

30 Mar 2023

PONE-D-23-01878Clostridium butyricum improves cognitive dysfunction in ICV-STZ-induced Alzheimer's disease mice via suppressing TLR4 signaling pathway through the gut-brain axisPLOS ONE

Dear Dr. Zhang,

Thank you for submitting your manuscript to PLOS ONE. After careful consideration, we feel that it has merit but does not fully meet PLOS ONE’s publication criteria as it currently stands. Therefore, we invite you to submit a revised version of the manuscript that addresses the points raised during the review process.

We look forward to receiving your revised manuscript.

Kind regards,

Yasmina Abd‐Elhakim

Academic Editor

PLOS ONE

Journal Requirements:

“The work had been funded by research start-up funds (00104311-2022-1-1-26), Program for Innovative Research Team (in Science and Technology) in University of Henan Province (21IRTSTHN026), ‘liangshan’ research project in Lishui University (FGL202202, FGL202204).”

“The work had been funded by research start-up funds (00104311-2022-1-1-26), Program for Innovative Research Team (in Science and Technology) in University of Henan Province (21IRTSTHN026), ‘liangshan’ research project in Lishui University (FGL202202, FGL202204).”

“The work had been funded by research start-up funds (00104311-2022-1-1-26), Program for Innovative Research Team (in Science and Technology) in University of Henan Province (21IRTSTHN026), ‘liangshan’ research project in Lishui University (FGL202202, FGL202204).”

7. In your Data Availability statement, you have not specified where the minimal data set underlying the results described in your manuscript can be found. PLOS defines a study's minimal data set as the underlying data used to reach the conclusions drawn in the manuscript and any additional data required to replicate the reported study findings in their entirety. All PLOS journals require that the minimal data set be made fully available. For more information about our data policy, please see http://journals.plos.org/plosone/s/data-availability.

Reviewers' comments:

Reviewer's Responses to Questions

**Comments to the Author**

1. Is the manuscript technically sound, and do the data support the conclusions?

Reviewer #1: Yes

Reviewer #2: Yes

Reviewer #3: Yes

2. Has the statistical analysis been performed appropriately and rigorously? 

Reviewer #1: Yes

Reviewer #2: Yes

Reviewer #3: Yes

3. Have the authors made all data underlying the findings in their manuscript fully available?

Reviewer #1: Yes

Reviewer #2: Yes

Reviewer #3: Yes

4. Is the manuscript presented in an intelligible fashion and written in standard English?

Reviewer #1: Yes

Reviewer #2: Yes

Reviewer #3: Yes

5. Review Comments to the Author

**Reviewer #1:** The manuscript entitled ‘Clostridium butyricum improves cognitive dysfunction in ICV-STZ-induced Alzheimer's disease mice via suppressing TLR4 signaling pathway through the gut-brain axis’ is fairly organized, well-written and the results are very interesting. The manuscript just needs English editing with native speaker.

**Reviewer #2:** I thank the authors for their tedious work and like to recommend the use of levodopa in addition to pre/probiotics in a third comparison group to study the interaction of both treatment modalities. Thank you.

**Reviewer #3:** "Clostridium butyricum improves cognitive dysfunction in ICV-STZ-induced Alzheimer's disease mice via suppressing TLR4 signaling pathway through the gut-brain axis"

The topic is interesting and study design is sound and well-conducted. The manuscript is organized and well-written, and may be suitable for publication after minor changes.

It is suggested to add a list of abbreviation to help understanding of the lots of abbreviated terms in the text.

Also, the authors, in page 4, lines 106 & 107, stated that "The right ventricle was slowly injected with STZ

107 (3 mg·kg-1) for 5 min", but in many other parts they stated that injected STZ in the lateral ventricle. please revise this sentence.

6. PLOS authors have the option to publish the peer review history of their article (what does this mean?). If published, this will include your full peer review and any attached files.

Reviewer #1: No

Reviewer #2: **Yes: **Mohamed Mostafa

Reviewer #3: No

---

## [Author Response · Author response to Decision Letter 0]

6 May 2023

Response to Editor

1.Response: Thank you for your suggestion. We revised the manuscript according to the PLOS ONE's style requirements.

2.Response: We correct the Funding Statement and list them in the cover letter. Thank you.

3.Response: We remove the Funding Statement from the Acknowledgments section of the manuscript and list them in the cover letter, thank you.

4.Response: We set up an ORCID iD, thank you.

5.Response: We amend the abstract on the online submission form, thank you.

6.Response: We add the original uncropped and unadjusted images in a PDF and upload it for a supplement file.

7.Response: The original contributions presented in the study are all included in the article/Supplementary Material, further inquiries can be directed to the corresponding author.

8.Response: There is no retracted reference in the manuscript text.

Response to Reviewers

Dear editors and reviewers:

The authors are grateful to the editors and reviewers for the comments and opportunity to revise the manuscript. The authors responded carefully to the comments and the manuscript was revised. Please give another opportunity to revise again if the editors and reviewers have different opinions or suggestions. Thank you.

Here are the comments and responses: 

Reviewer #1: The manuscript entitled ‘Clostridium butyricum improves cognitive dysfunction in ICV-STZ-induced Alzheimer's disease mice via suppressing TLR4 signaling pathway through the gut-brain axis’ is fairly organized, well-written and the results are very interesting. The manuscript just needs English editing with native speaker.

Response: We agree with this suggestion. The manuscript was edited by an English native speaker.

Reviewer #2: I thank the authors for their tedious work and like to recommend the use of levodopa in addition to pre/probiotics in a third comparison group to study the interaction of both treatment modalities. Thank you.

Response: Thank you for your suggestion. This study will be better explained if there is a third comparison group to study the interaction of both treatment modalities. In the future, a comparison group with both levodopa and pre/probiotics will be added and the interaction of them will be tested.

Reviewer #3: "Clostridium butyricum improves cognitive dysfunction in ICV-STZ-induced Alzheimer's disease mice via suppressing TLR4 signaling pathway through the gut-brain axis"

The topic is interesting and study design is sound and well-conducted. The manuscript is organized and well-written, and may be suitable for publication after minor changes.

It is suggested to add a list of abbreviation to help understanding of the lots of abbreviated terms in the text.

Also, the authors, in page 4, lines 106 & 107, stated that "The right ventricle was slowly injected with STZ

107 (3 mg·kg-1) for 5 min", but in many other parts they stated that injected STZ in the lateral ventricle. please revise this sentence.

Response: Thank you for your suggestion. (1) A list of abbreviation was added in the manuscript (Please see Table 1). (2) The sentence of "The right ventricle was slowly injected with STZ (3 mg·kg-1) for 5 min" was revised to “The right lateral ventricle was slowly injected with STZ (3 mg·kg-1) for 5 min”. Please see the revised manuscript. (lines 110 in the manuscript)

Sincerely Yours,

Zhenqiang Zhang

Henan Engineering Research Center for Prevention and Treatment of Major Chronic Diseases with Chinese Medicine

Academy of Chinese Medical Sciences

Henan University of Chinese Medicine

156 Jinshui Dong Road

450046 Zhengzhou

Henan province, PR China

Email: 13333719963@126.com

Tel: 13333719963

Date: 6-May-2023

---

## [Editor Report · Decision Letter 1]

9 May 2023

Clostridium butyricum improves cognitive dysfunction in ICV-STZ-induced Alzheimer's disease mice via suppressing TLR4 signaling pathway through the gut-brain axis

PONE-D-23-01878R1

Dear Dr. Zhang,

We’re pleased to inform you that your manuscript has been judged scientifically suitable for publication and will be formally accepted for publication once it meets all outstanding technical requirements.

Kind regards,

Yasmina Abd‐Elhakim

Academic Editor

PLOS ONE
---

## [Editor Report · Acceptance letter]

24 May 2023

PONE-D-23-01878R1 

*Clostridium butyricum* improves cognitive dysfunction in ICV-STZ-induced Alzheimer's disease mice via suppressing TLR4 signaling pathway through the gut-brain axis 

Dear Dr. Zhang:

I'm pleased to inform you that your manuscript has been deemed suitable for publication in PLOS ONE. Congratulations! Your manuscript is now with our production department. 

Kind regards, 

on behalf of

Prof. Dr. Yasmina Abd‐Elhakim 

Academic Editor

PLOS ONE